# A Study on the Spatial Structure of the Bu-Ul-Gyeong Megacity Using the City Network Paradigm

**Yoonjee Baek [1] and Heesun Joo [2],***

1   Department of Urban Planning and Design, School of Architecture, Tsinghua University, Beijing 100084, China
2   Department of Urban Engineering, Gyeongsang National University, Jinju 52828, Republic of Korea
*   Correspondence: hsjoo@gnu.ac.kr; Tel.: +82-055-772-1771

**Abstract:** Developing bidirectional urban networks within areas in megacities is an essential spatial strategy across regions today. In 2018, Korea began its Bu-Ul-Gyeong (BUG) megacity project. Today, Korea is working to improve functional polycentric urban networks within the BUG megacity. To uncover insights useful for this project, this study sought to examine urban network patterns (e.g., network asymmetries and imbalances in the sizes and directions of their weighted flows) and identify the primary and secondary centers of the BUG megacity using mobile flow data from 2019 to 2020. Specifically, a three-step social network analysis was conducted across different geographical scales; namely: (1) the BUG megacity, (2) South Gyeongsang Province (SGP), and (3) every community in SGP. Eigenvector centrality and flow betweenness centrality revealed two primary centers (Changwon and Jinju) and four secondary centers (Haman, Sacheon, Tongyeong, and Geochang). Unidirectional and hierarchical connections were evident between the primary and secondary centers. In response to these findings, this paper proposes some beneficial strategies for the region's public transportation networks to prevent small- and medium-sized cities from being marginalized and to enhance horizontal urban connectivity in megacities.

**Keywords:** city network paradigm; social network analysis; centrality; urban spatial structure; megacity; central city





## 1. Introduction

Over two-thirds of the world's population and wealth are concentrated in megacities [1]. The United Nations defines a "megacity" as a city with over ten million inhabitants and estimates that 43 megacities will exist across the world by 2030 [2]. Thinkers have used various terms to capture the style of urban agglomeration and regional integration characteristic of what the United Nations terms a megacity, such as "mega-city region" in Europe [3], "city-cluster region" in China [4], and "megalopolis" in the United States [5]. Regardless of terminology, this style of regional integration can be further divided into "domestic" and "transnational" integration. The former involves building urban networks between cities and counties to create an integrated economic community. The latter involves the creation of a political and economic union, such as the Association of Southeast Asian Nations (ASEAN) and the European Union (EU) [6]. Notably, domestic integration is often celebrated for its ability to cultivate stability and global competitiveness [7]. This study focused on domestic integration by exploring the case of the Bu-Ul-Gyeong (BUG) megacity project in South Korea (this paper uses the term "megacity" to indicate a geography containing multiple metropolitan areas, cities of different sizes, and neighboring rural areas).

Before turning to the case at hand, a summary of existing research and practice related to the topic of megacities is necessary. Recently, the city network paradigm has emerged as a notable new framework for considering the urban spatial structure of megacities [8]. Many studies have already investigated the spatial structures of diverse aspects of megacities

based on the city network paradigm; for example, work has already been done on European megacities [9] and global megacities [10,11]. Moreover, some studies have analyzed communities in integrated regions based on the city network theory; for example, such analyses have been done using commuting data from megacities in the United States [12,13]. Meanwhile, Zhang and Lan [14] examined the spatial patterns of megacities in the United States using commuting flow data from 2011 to 2015. Their study included a two-step analysis: first, they conducted a national-level study; second, they conducted a state-level study (of Texas) using the weighted community detection algorithm. Ultimately, they found that urban places (e.g., Houston and Dallas, the Northeast and Florida) can be integrated into one cluster to overcome proximity constraints.

To date, many countries have focused on how to manage megacities and apply the spatial structure of a network city as an essential national spatial strategy. In particular, megacity planning and construction have become significant parts of South Korea's national strategies for balanced national development. Today, over half of Korea's population lives in its metropolitan areas [15]. Consequently, non-metropolitan areas are facing problems such as a demographic cliff, regional economic stagnation, and the risk of extinction [16] (pp. 3–9). These issues have prompted discussions on hyper-regionalization strategies, such as the formation of megacities in non-metropolitan areas to build economic communities and balanced regional development. The creation of functional polycentric urban networks can promote spatially balanced development [17]. Accordingly, the fundamental purpose of such strategies is to change Korea's spatial structure from monocentric to polycentric. The polycentric system stresses diverse horizontal and bidirectional networks among regions; crosses the boundaries of administrative districts; and encourages industry revitalization, population growth, infrastructure development, and improved urban living environments. Further, the spatial structure of a network city highlights the need for separate but harmonious and cooperative urban functions, such as administration, industry, culture, and welfare [18].

The BUG megacity project was established in this vein in 2018. The BUG megacity includes Busan, Ulsan, and South Gyeongsang Province (hereafter SGP). Notably, it aims to become an economically and culturally integrated community. The spatial structure plan designates the four cities of Busan, Ulsan, Changwon, and Jinju as the megacity's central cities. Each city is expected to perform significant roles by integrating networks in surrounding small- and medium-sized cities. The urban linkages among the areas are recommended to be horizontal rather than hierarchical. However, the area's existing spatial structure is limited in that it has scarcely defined boundaries. As a result, existing plans do not adequately address how the major areas should be connected. In response to this gap, this study primarily sought to uncover information useful for determining the best practices for connections between different parts of the BUG megacity. Specifically, this study sought to identify which cities are functioning as central cities in the current urban network patterns and the primary and secondary centers of the BUG megacity by applying city network theory. To analyze the urban network on macroscopic and microscopic scales, this study carried out a social network analysis across three different geographical scales: (1) the BUG megacity, (2) the SGP, and (3) every community in the SGP.

The rest of this paper is organized as follows. Section 2 reviews the literature on city network theory, spatial structure, urban regions, and primary and secondary centers. Section 3 outlines the research setting, data, and methodology. Section 4 reports the analysis results. Section 5 discusses the major findings and some practical spatial strategies.

## 2. Literature Review

### 2.1. City Networks and Spatial Structure

Cappello [19] defined the "city network paradigm" as the network in which cities can achieve economies of scale through complementary interactions, synergies, and collaborative actions (p. 1925). The network theory can observe horizontal relationships between cities and is thus a proper theoretical framework to overcome the limitations of the central

place model [19]. The city network paradigm has been adopted to articulate the polycentric spatial structure of cities, urban regions, and international cities [20].

Since the second half of the 20th century, scholars have applied the concept of a city network paradigm to urban regions, including megacities [21], mega-regions [22], megacity regions [23], and global cities [24]. Castells [21] defined "megacities" as the hubs of a global economy and urban agglomerations [25]. Florida et al. [22] clarified that mega-regions have considerable economic power across administrative units. Meanwhile, Hall and Pain [23] argued that megacities are spatially divided into several cities but functionally linked to maximize their economic values by focusing on a central area.

Taylor et al. [26] identified the two polycentric city-regional processes in cities and towns in the UK: a polycentric mega-city regional process and a polycentric multi-city regional process. The former involves the creation of a new central area for urban services and the provision of services to selected neighboring small cities and towns. The latter mostly highlights the selected large cities' service capacities. The city network paradigm emphasizes both the linkages between the areas within an urban region and the functional interactions among areas by forming a polycentric spatial structure [23]. Westin and Östhol [27] stated that the city network involves cooperation and collaborative governance among neighboring spatial units and the efficient division of the units' functions [28] to generate synergy in urban regions [29]. Batten [30] remarked that the city network is a system that allows multiple individual cities to functionally amplify cooperation, emphasizes the nodality rather than the centrality of urban regions, and shares benefits by reducing transaction costs. Additionally, Eliasson et al. [31] pointed out that city networks can serve as spatial strategies to promote intra-industry trade and create incentives for economic cooperation. Van Oort et al. [32] observed hierarchies in the different types of spatial interdependencies (The types of networks within the same urban region are as follows (Figure 1): intra-nodal (Type 1), core–periphery (Type 2), and criss-cross (Type 3). The types of networks between urban regions are as follows (Figure 1): inter-core (Type 4), core–periphery (Type 5), and criss-cross (Type 6) [32] (p. 738)). in the Netherlands' Randstad city network and defined several networks between urban regions: inter-core (Figure 1, Type 4), core–periphery, and criss-cross (Figure 1, Types 5, 6) [25,33].

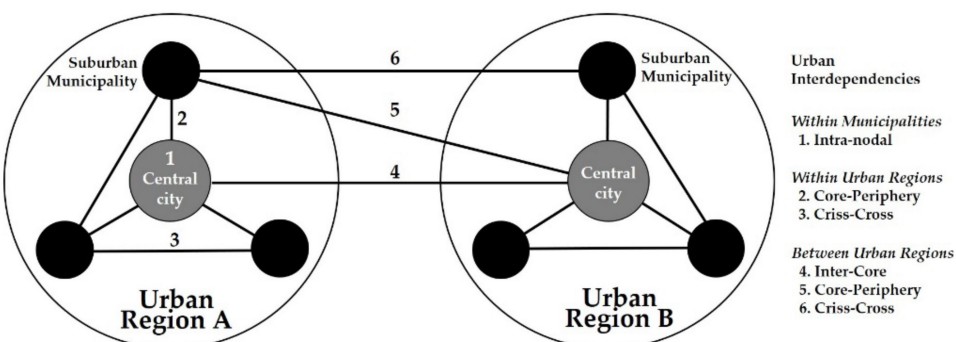

**Figure 1.** Classification of different types of urban interdependencies. Source: readapted with permission from [32]. 2010, Van Oort et al.

The urban spatial structure of a city network ("Urban spatial structure" can be defined as the pattern of spatial distribution resulting from interactions among urban elements such as land use and buildings, urban activities [34], urban networks [35]). highlights the horizontal and complementary integration among urban regions based on creating transportation networks and supplementing roles among cities [19,30]. Additionally, when multiple central areas are growing, it underlines not hierarchical order but horizontal accessibility among the central areas. It stresses nodality in city networks, such as traffic and communication networks, to identify the urban spatial structure and evaluates urban potentiality as exchanging information and its cost [30,36]. The city network paradigm is in the same vein as the functional perspective of a polycentric spatial structure in that

it emphasizes multiple central areas' linkages and interactions in urban regions. Both Green [37] and Meijers [38] argued that the morphological and functional dimensions of spatial structures should be considered to understand polycentric spatial structures. In particular, the functional dimension highlights connectivity among central areas [39]. In a functional polycentric spatial structure, the central areas should be networked in a two-way directional flow rather than in a unidirectional flow (Figure 2).

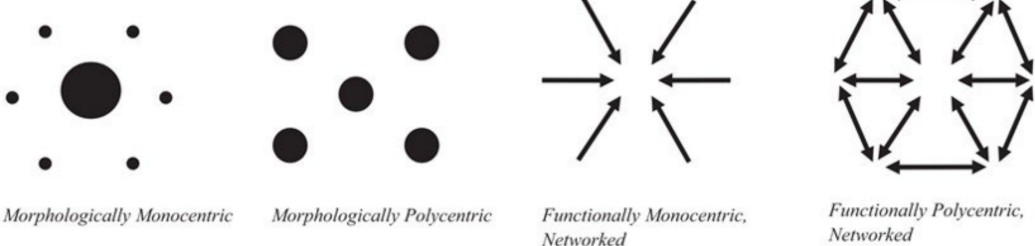

**Figure 2.** Morphologically versus functionally polycentric. Source: readapted with permission from [40]. 2012, Burger and Meijers.

## 2.2. Urban Regions

Previous papers clarify how urban regions (characterized by areas sharing similar attributes) have adopted the travel-to-work areas (TTWAs) or local labor market areas (LLMAs) methodologies to define themselves [41]. However, considerable research largely explores urban regions based on their functional dimensions [42], such as their relative functional polycentricity. "Functional polycentricity" is a characteristic of a region in which areas are linked across diverse functions. Therefore, cities within a polycentric urban region demonstrate functional interconnectedness and tend to be of similar size [37]; in this way, they stand in contrast to cities with a hierarchical spatial distribution [43].

Most studies examining urban regions in Korea have analyzed flow data and limited their geographical scope to the Korean peninsula [44–47]. Byun et al. [47] surveyed existing methodologies for exploring urban regions and stated the institutional and political implications for realizing balanced regional development. Some studies have analyzed the spatial structure of the southeast region of Korea, including Busan, Ulsan, and SGP [28,48]. Kwon [28] observed the spatial structure of the southeast region using indicators such as industrial clusters and population growth rates and articulated that the urban region followed the spatial structure of the city network model in that the growth rates of small- and medium-sized cities were high. In addition, the cities within the urban regions were mutually independent and closely connected through a well-established transport network and functioned in different roles. To measure and define the urban regions' community detection, one method of social network analysis, was used [49–52]. Kwon [42] analyzed the central areas using both the method of local labor market areas (LLMAs) and community detection and advised that community detection, which limits the researcher's involvement, is the best methodology for analyzing polycentric spatial structures.

## 2.3. Analysis of Primary and Secondary Centers

Several studies have addressed the polycentric spatial structure of cities or urban regions, identifying their primary centers and hierarchical relationships [53–56]. The way to define primary centers is relatively uncontentious. However, diverse measures were taken to observe secondary centers [53]. The traditional approach to analyzing the secondary centers was to use a cut-off point for employment density [53–56]. This would be in the same vein as studies focusing on the morphological dimensions of polycentric spatial structures. This research estimated secondary centers using the following indicators: employment density and number of employees [57–61], the traffic volume of every industry [62], the areas of office buildings [63–65], and land prices [60,61,66].

Much of the recent literature has highlighted the role of central cities, emphasizing the functional dimension of polycentric spatial structures or describing spatial structures using city network theory. Sohn [25] followed the attributes of the network city model suggested by Batten [30]: heterogeneous services, complementarity, and two-way flows. This study [25] identified the spatial structure of the Seoul metropolitan area based on three indicators: inequalities in the population distribution; employment rate of urban industry; and population rate of commuting. As a result, it asserted that the metropolitan area had a hierarchical spatial structure rather than the spatial pattern of the network city model. Other studies [67,68] used social network analysis with flow data to evaluate secondary centers. The flow data could be specified by purposes of trips such as commuting, returning home, shopping, and leisure [51,69–73], or types of transportation, such as the bus, the subway, or a personal vehicle [73,74]. As an outcome of the social network analysis, several centrality indices (e.g., eigenvector centrality and betweenness centrality) were used to evaluate secondary centers. Recent studies considered the following indices: degree centrality [75]; eigenvector centrality [51,76]; both degree centrality and eigenvector centrality [71,73]; and degree centrality, eigenvector centrality, and closeness centrality [52]. Different indices were used to examine the central cities, while the degree, betweenness, and eigenvector centralities were commonly adopted.

To summarize the key takeaways from the literature review so far, most existing studies have addressed the Seoul metropolitan area to investigate its urban centers and spatial structure. However, few in-depth studies have been conducted focusing on the southeastern region of Korea, that is, the BUG megacity. In addition, previous studies analyzing central cities have not adequately addressed the hierarchy of the central cities. The purpose of this paper is to fill this research gap. Hence, this study clarified the spatial structure of the BUG megacity and investigated the primary and secondary centers by conducting a network analysis using mobile flow data.

## 3. Research Design

### 3.1. Research Area

The geographical scope of this study was the BUG megacity—comprising Busan metropolitan city, Ulsan metropolitan city, and South Gyeongsang Province (SGP) (including eight cities and ten counties)—located in the southeastern region of Korea (Figure 3). The Korean government first began the BUG megacity project in the early 2000s; however, at that time it faced many obstacles due to a lack of institutional support. To overcome these difficulties, the Local Autonomy Act was revised in January 2022 [77], which created the conditions necessary to establish a special coalition self-governing body (special SGB) to extend multiple cities and municipalities. At this time, the BUG megacity's population was approximately 7.78 million, accounting for 0.15% of the total population of Korea. In April 2022, the BUG special SGB was established as Korea's first special SGB [78]. Notably, the BUG megacity project involves experts from various economic, administrative, and sociocultural fields dedicated to creating a unified urban region [16]. To date, work to economically and culturally integrate the region has emphasized the importance of enhancing inter-regional transportation networks within an hour of the BUG megacity [79]. In light of the recent legislation, the BUG special SGB could request and operate plans related to public transportation and bus rapid transit (BRT) systems across regional borders under the authority of the central government Ministry of Land, Infrastructure and Transport [80].

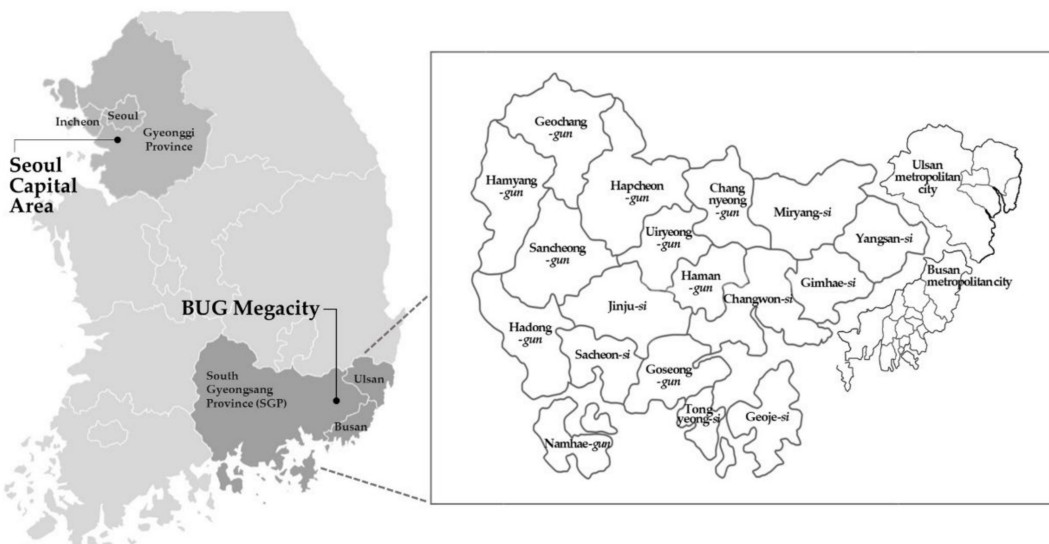

**Figure 3.** Location of the BUG megacity.

*3.2. Data Sources*

This study used the mobility data from mobile phones collected by SKT (Korea's three major mobile carriers and their share of their subscribers as of 2020 were SKT (47.1%), KT (29.7%), and LG U+ (24%) [81]. SKT Telecom had the highest market share of the three; thus, it would reflect the mobility patterns properly. Data were from the total population and were collected by SKT base stations every fifteen minutes. According to the length of stay at a specific location and time zone, the data were divided into the resident population, office worker population, and visiting population). for a total of four weeks: 18–24 March 2019, 23–29 September 2019, 18–24 March 2020, and 21–27 September 2020. Data on the total population, including those residing, working, and visiting in the region, were used to identify the overall population mobility patterns within the BUG megacity. Moreover, the study utilized the average daily population flow for four weeks.

Lee et al. [75] urged that failing to consider the size of the origin and destination areas could over- or under-estimate the result when analyzing an area's centrality to investigate population mobility patterns. They advised that the local population should be considered and that "population flow per local population" should be used to analyze the area's centrality to minimize this problem [75]. Based on the abovementioned measure, this study considered the local population by using data on the resident registration population provided by the Ministry of Interior and Safety in March and September 2019, 2020. Specifically, this study used "population flow per 100 local population" (P.F./100 L.P.) data from Busan, Ulsan, and SGP. This denotes the number of people who traveled daily from the origin (O) to the destination (D) per every 100 people in the O's local population.

*3.3. Research Method*

3.3.1. Social Network Analysis and Investigating Primary and Secondary Centers

This study examined the spatial structure and central cities of the BUG megacity by conducting a social network analysis. Social network analysis explores the relationships among nodes, including groups and services as cited in [82,83]. Based on graph theory, each object is considered a node in social network analysis; it analyzes the numerical flows among nodes cited in [84,85]. In this study, the nodes were deemed as twenty areas, indicating two metropolitan cities, eight cities, and ten counties in the BUG megacity; the flows were regarded as the P.F./100 L.P. We performed a centrality analysis to measure the centrality of each node. Typical centrality indicators include degree, closeness, betweenness, and eigenvector. Among them, this study used eigenvector and flow betweenness centrality to identify the central cities, focusing on horizontal relations and complementary interaction

among areas. The centrality analysis was performed by the social network analysis program for Netminer version 4.0. As a result, we acquired the values of eigenvector centrality and flow betweenness centrality in each of the twenty areas. The areas that ranked within the top 20% for both centrality indicators were defined as the central cities (Roh et al. [86] used the criterion for defining the central cities as above Z-score value of 0.8. However, the flow data in this research follow a non-normal distribution; therefore, it was hard to apply the Z-score value. Kim and An [61] regarded the above Z-score value of 0.8 as the top 20%. By following the study [61], this study also considered the criterion for examining the central cities as the areas in the top 20%. SGP contained a total of eighteen cities and counties and the top 20% areas were 3.6. We deemed the top four areas the central cities).

(1) Eigenvector Centrality

*Eigenvector centrality* (EC) applies Bonacich's methodology [87] and measures a node's transitive influence. The EC is calculated as per Equation (1), where $x$ denotes the corresponding principal eigenvector and $\lambda$ denotes the largest eigenvalue. The EC $C_i$ of a node $i$ is defined by the sum of the values within the principal eigenvector $C$ corresponding to direct neighbors, as defined by the adjacency matrix (i.e., where $A_{ij} \neq 0$). Subsequently, the EC is scaled by the proportionality factor $1/\lambda$ [88].

$$C_i = \frac{1}{\lambda} \sum_{j=1}^{n} A_{ij} C_j \tag{1}$$

The EC value is often used as an indicator to identify central cities [51,71,73,75,76] because it presents the degree of influence on all connections within the areas under study [81]. When calculating the EC value, relations originating from high-scored nodes contribute more to the score of a node than connections from low-scored nodes. A high EC score means that a node is linked to many nodes that themselves have a high influence on the urban network [89].

(2) Flow Betweenness Centrality

*Flow betweenness centrality* (hereafter FBC) measures how involved the node is in all the flows between all other pairs of nodes, as a ratio of the total flow betweenness that does not involve the node. If a network has $N$ nodes, then the flow betweenness of node $i$ can be calculated as per Equation (2), where $C_B(i)$ is the flow betweenness of node $i$, $A_{jk}$ is the total count of all possible paths between node $j$ and node $k$, and $A_{jk}(i)$ is the total count of all possible paths between node $j$ and node $k$ passing through node $i$ [90].

$$C_B(i) = \sum_{j<k} \frac{A_{jk}(i)}{A_{jk}} \tag{2}$$

The FBC value of area A is defined as follows: 1- (total amount of flows between two areas when area A is excluded from the network/total amount of flows between two areas) [89]. An area that has a high value of FBC is regarded as a significant mediator in the urban network [91,92]. Betweenness centrality (BC) is a similar indicator to the FBC, which calculates the number of shortest paths between two areas. Even if an area could not have the many shortest connections to other areas, areas having a high level of FBC could be regarded as the most essential mediator in the overall network [30,67]. This study aimed to identify the primary and secondary centers of SGP considering the overall urban network of SGP. Therefore, the value of BC was adopted.

### 3.3.2. Community Detection

Community detection is a method to find a particular division of networks. In urban networks, several groups or clusters could be detected and they have sets of nodes with a high density of internal flows, while flows between groups have a relatively low density. These groups are deemed "modules" or "communities" [93]. Among the diverse algorithms of community detection, this study used the algorithm proposed by Blondel et al. [94]. This

algorithm follows a local optimization method suggested by Girvan and Newman [93] to investigate some partitions. Based on this, modules are exchanged by supreme nodes and produce a lower density in the small network. This step is iterated until the value of modularity no longer increases. The anymore modularity ranges from −1 to 1 [94]. The algorithm is available in the network analysis and visualization program for Gephi; thus, the community detection was performed by Gephi version 0.9.7.

### 3.4. Structure of Research

This study aimed to investigate the spatial structure of the BUG megacity and clarify its primary and secondary centers based on city network theory [32,33]. As noted above, to determine the primary and secondary centers, it performed a three-step urban network analysis across different geographical scales: (1) the BUG megacity, (2) the SGP, and (3) every community in the SGP. This study considered area centrality across multi-level scales because area centrality can be differently defined depending on the geographical scale. The first step of the network analysis involved analyzing the urban networks among Busan, Ulsan, and SGP. Specifically, external (The external flow of SGP indicates the network among the 18 cities and counties of SGP, Busan, and Ulsan. It was difficult to use population mobility data between Busan and Ulsan; therefore, we examined outflows/inflows between SGP and Busan and SGP and Ulsan). origin-destination (OD) matrices were estimated and network analysis was conducted. The second step was focused on the SGP; thus, internal (The internal flow of SGP represents the network among the 18 cities and counties of SGP). origin–destination (OD) matrices were calculated and network analysis was conducted. Next, the primary centers were identified based on the results of the second step of the network analysis. The third step involved a network analysis of every community in the SGP. This was necessary to explore the communities within the SGP in advance; therefore, community detection was performed first. Based on the outcome of the community detection, a network analysis was conducted on each community in the SGP. In turn, the secondary centers were identified in each of the communities.

## 4. Results

### 4.1. External Flow of SGP

#### 4.1.1. In/Outflow Pattern

The origin–destination (OD) matrix was estimated based on the value of P.F./100 L.P. from eighteen areas of SGP(O) to Busan(D) and Ulsan(D) (Table 1). Firstly, the P.F./100 L.P. from SGP to Busan, Yangsan (15.38), Gimhae (11.57), Miryang (4.9), and Geoje (4.05) showed a high weight of population mobility. In particular, the flows from Yangsan and Gimhae to Busan were extremely high. Secondly, the population flow pattern from SGP to Ulsan, Yangsan (4.78), Miryang (0.97), Gimhae (0.59), and Geoje (0.39) presented a high volume of population mobility, which was relatively lower than those of Busan. To summarize, outstanding outflows were observed from four areas of SGP: Yangsan, Gimhae, Miryang, and Geoje. These were geographically close to Busan and Ulsan.

Furthermore, the origin–destination (OD) matrix was calculated using the index of P.F./100 L.P. from Busan(O) and Ulsan(O) to eighteen areas of SGP(D) (Table 2). Firstly, for the inflow pattern from Busan, Yangsan (221.87) had the highest volume, followed by Gimhae (216.52) and Changwon (131.74). It showed that the population inflow pattern to Yangsan, Gimhae, and Changwon from Busan was particularly outstanding. Secondly, for the inflow pattern from Ulsan, Yangsan (57.10), Changwon (21.33), and Gimhae (16.15) showed a high ranking within the eighteen areas of SGP. However, this inflow pattern was comparatively weaker than that of Busan. To sum up, the inflows from Busan and Ulsan to Yangsan, Gimhae, and Changwon were remarkable, and the three areas were spatially close to Busan and Ulsan.

**Table 1.** SGP(O)-Busan and Ulsan(D) matrix.

| | Origin | | Destination | |
|---|---|---|---|---|
| **No** | **SGP** | | **Busan** | **Ulsan** |
| 1 | Yangsan | city (*si*) | 15.38 | 4.78 |
| 2 | Gimhae | city (*si*) | 11.57 | 0.59 |
| 3 | Miryang | city (*si*) | 4.90 | 0.97 |
| 4 | Geoje | city (*si*) | 4.05 | 0.39 |
| 5 | Changwon | city (*si*) | 3.94 | 0.32 |
| 6 | Uiryeong | county (*gun*) | 2.65 | 0.3 |
| 7 | Haman | county (*gun*) | 2.59 | 0.31 |
| 8 | Changnyeong | county (*gun*) | 2.40 | 0.28 |
| 9 | Namhae | county (*gun*) | 2.39 | 0.18 |
| 10 | Goseong | county (*gun*) | 2.33 | 0.33 |
| 11 | Tongyeong | city (*si*) | 2.02 | 0.2 |
| 12 | Hadong | county (*gun*) | 1.90 | 0.2 |
| 13 | Sancheong | county (*gun*) | 1.66 | 0.18 |
| 14 | Hapcheon | county (*gun*) | 1.62 | 0.2 |
| 15 | Sacheon | city (*si*) | 1.55 | 0.22 |
| 16 | Hamyang | county (*gun*) | 1.37 | 0.15 |
| 17 | Jinju | city (*si*) | 1.32 | 0.18 |
| 18 | Geochang | county (*gun*) | 0.92 | 0.19 |

The value of P.F./100 L.P: the daily number of the population who travel from the origin to destination per origin's 100 local population.

**Table 2.** Busan and Ulsan(O)- SGP(D) matrix.

| Origin | Yangsan City (*si*) | Gimhae City (*si*) | Changwon City (*si*) | Geoje City (*si*) | Miryang City (*si*) | Jinju City (*si*) | Tongyeong City (*si*) | Haman County (*gun*) | Sacheon City (*si*) |
|---|---|---|---|---|---|---|---|---|---|
| | | | | | **Destination** | | | | |
| Busan | 221.87 | 216.52 | 131.74 | 42.91 | 34.96 | 21.22 | 18.28 | 11.57 | 10.84 |
| Ulsan | 57.10 | 16.15 | 21.33 | 8.58 | 11.75 | 6.07 | 5.43 | 2.30 | 3.04 |

| Origin | Namhae County (*gun*) | Changnyeong County (*gun*) | Goseong County (*gun*) | Hadong County (*gun*) | Sancheong County (*gun*) | Hapcheon County (*gun*) | Uiryeong County (*gun*) | Hamyang County (*gun*) | Geochang County (*gun*) |
|---|---|---|---|---|---|---|---|---|---|
| | | | | | **Destination** | | | | |
| Busan | 10.40 | 10.04 | 8.83 | 8.35 | 6.67 | 5.87 | 5.66 | 4.05 | 3.80 |
| Ulsan | 2.59 | 2.18 | 2.23 | 2.12 | 1.85 | 1.81 | 0.84 | 1.25 | 1.50 |

The value of P.F./100 L.P: the daily number of the population who travel from the origin to destination per origin's 100 local population.

This section investigated population flows among SGP, Busan, and Ulsan using OD matrices. The outcome results of analyzing the external flows of SGP identified that Busan and SGP had stronger networks rather than those between Ulsan and SGP. In the outflow pattern, Yangsan, Gimhae, and Miryang showed powerful networks with Busan and Ulsan; for the inflow pattern, Yangsan, Gimhae, and Changwon had distinct linkages. In other words, Yangsan and Gimhae were strongly connected with Busan and Ulsan for both inflows and outflows. Changwon had a more powerful inflow pattern than the out-flows. It needed to be clarified more to determine the central cities in the external network of SGP; thus, network analysis was performed in the following part.

4.1.2. Network Analysis

The network analysis was conducted using the external network of SGP, including flows within eighteen areas in SGP, Busan, and Ulsan. This study revealed central cities of SGP were ranked in the top 20% among eighteen areas based on the results of network analysis. As a result, Yangsan, Gimhae, and Geoje had high values of both EC and FBC (Table 3, Figure 4). As mentioned above, in their value of P.F./100 L.P., Yangsan and Gimhae were highly interconnected with Busan and Ulsan in both directions. Moreover, the two cities had high positions in the hierarchy and functioned as mediators in the

external networks of SGP. In addition, Gimhae and Geoje had significant influences on the external network and performed a vital mediating role. Following the result, Yangsan, Gimhae, and Geoje would be considered the central cities of the external networks of SGP.

**Table 3.** Results of network analysis of the external flow of SGP.

| No | City/County | EC | City/County | FBC |
|---|---|---|---|---|
| 1 | Busan | 0.696 | Busan | 1393.92 |
| 2 | Yangsan | 0.464 | Ulsan | 235.70 |
| 3 | Gimhae | 0.439 | Yangsan | 76.32 |
| 4 | Changwon | 0.271 | Gimhae | 37.66 |
| 5 | Ulsan | 0.120 | Miryang | 25.42 |
| 6 | Geoje | 0.089 | Geoje | 18.93 |
| 7 | Miryang | 0.074 | Changwon | 13.74 |
| 8 | Jinju | 0.045 | Haman | 11.37 |
| 9 | Tongyeong | 0.038 | Goseong | 11.23 |
| ... | | ... | | ... |
| 18 | Uiryeong | 0.012 | Hamyang | 6.05 |
| 19 | Hamyang | 0.009 | Uiryeong | 5.46 |
| 20 | Geochang | 0.008 | Geochang | 5.01 |

Concentric map [1]

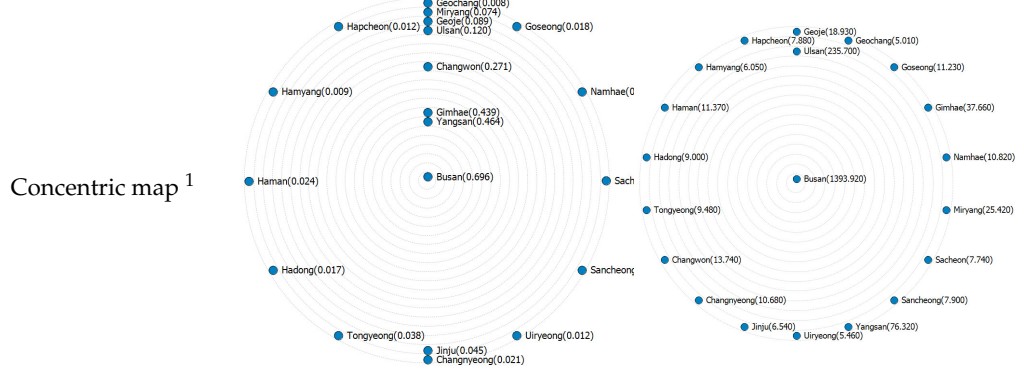

[1] In the concentric map, an area with a high value of centrality was located at the center of the circle.

In Figure 4, an area that had the higher value of centrality had a larger and darker circle. The arrows represented "population flow per 100 local population" (P.F./100 L.P.), which if it had the higher value, it had thicker and darker arrows. The pointing of arrows indicated the flow direction of the value.

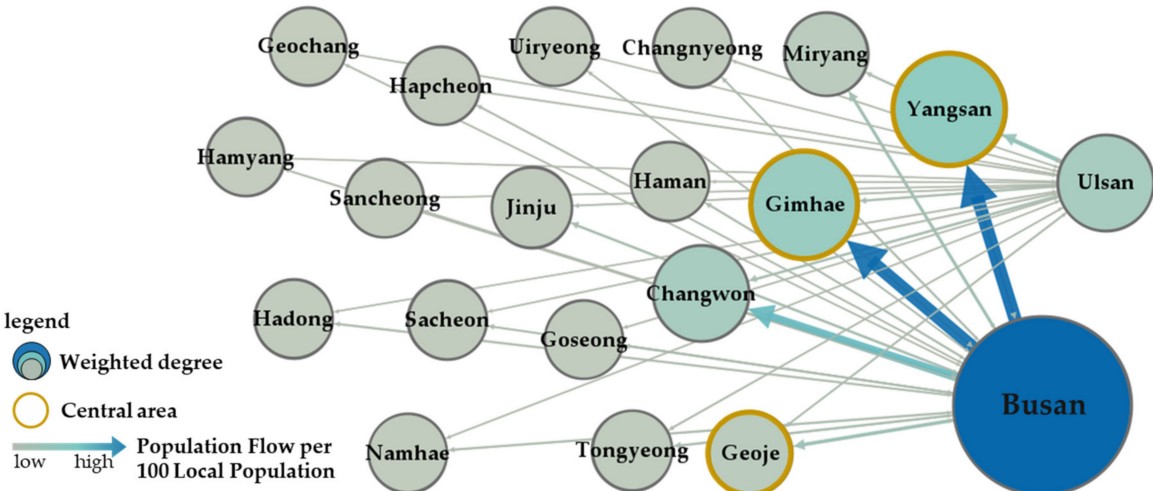

**Figure 4.** Urban networks and central areas of the external flow of SGP.

*4.2. Internal Flow of SGP*

4.2.1. In/Outflow Pattern

The OD matrix was generated using the value of P.F./100 L.P. within eighteen areas of SGP. In the P.F./100 L.P. (Table 4) within SGP, from Haman to Changwon (1532.2) had the most outstanding flow, followed by Sancheong–Jinju (651.5), Sacheon–Jinju (595.3), Tongyeong–Geoje (510.9), Gimhae–Changwon (508.6), and Uiryeong–Changwon (461.2). Overall, areas of SGP were highly connected with Changwon and Jinju in their inflow patterns as shown in the OD matrix (Table 4). Firstly, the next areas showed a high degree of population inflow to Changwon: Haman (1532.2), Gimhae (508.6), Uiryeong (461.2), Changnyeong (365.3), and Goseong (364.8). All five remarkable areas were close to Changwon and had stronger inflows to Changwon than outflows from Changwon. Secondly, the following areas presented a high level of population inflows to Jinju: Sancheong (651.5), Sacheon (595.3), Hadong (379.0), and Goseong (195.4). The four noticeable areas were not only geographically close to Jinju but also had higher inflows to Jinju than out-flows from Jinju. In summary, in the internal network of SGP, it was identified that the inflow patterns to Changwon and Jinju were highly prominent from the spatially adjacent areas. It was required to more specifically identify the central cities in the internal network of SGP; therefore, network analysis was conducted in the next part.

**Table 4.** OD matrix of internal flow of SGP.

| OD | 1 | 2 | 3 | 4 | 5 | 6 | 7 | 8 | 9 | 10 | 11 | 12 | 13 | 14 | 15 | 16 | 17 | 18 |
|----|---|---|---|---|---|---|---|---|---|----|----|----|----|----|----|----|----|----|
| 1 | - | 365 | 5 | 10 | 69 | 8 | 3 | 223 | 3 | 33 | 29 | 71 | 4 | 10 | 5 | 11 | 40 | 16 |
| 2 | 46 | - | 5 | 30 | 205 | 42 | 8 | 165 | 5 | 8 | 48 | 33 | 9 | 18 | 9 | 24 | 24 | 17 |
| 3 | 9 | 45 | - | 6 | 17 | 5 | 5 | 5 | 189 | 112 | 53 | 4 | 5 | 14 | 26 | 9 | 3 | 7 |
| 4 | 4 | 96 | 3 | - | 36 | 41 | 8 | 6 | 4 | 3 | 42 | 5 | 7 | 17 | 8 | 152 | 2 | 16 |
| 5 | 20 | 509 | 3 | 25 | - | 13 | 7 | 24 | 4 | 6 | 26 | 69 | 7 | 12 | 6 | 15 | 7 | 65 |
| 6 | 7 | 365 | 4 | 81 | 48 | - | 11 | 20 | 9 | 4 | 195 | 5 | 15 | 268 | 11 | 358 | 6 | 13 |
| 7 | 3 | 75 | 4 | 13 | 25 | 14 | - | 11 | 8 | 4 | 379 | 3 | 102 | 121 | 49 | 11 | 5 | 11 |
| 8 | 181 | 1532 | 6 | 20 | 91 | 29 | 9 | - | 6 | 13 | 83 | 16 | 10 | 24 | 10 | 24 | 124 | 17 |
| 9 | 3 | 50 | 306 | 8 | 20 | 7 | 8 | 7 | - | 14 | 181 | 3 | 5 | 20 | 143 | 11 | 6 | 11 |
| 10 | 46 | 78 | 97 | 9 | 23 | 7 | 4 | 16 | 8 | - | 139 | 5 | 4 | 15 | 33 | 8 | 53 | 12 |
| 11 | 6 | 118 | 17 | 23 | 30 | 63 | 95 | 36 | 38 | 44 | - | 6 | 47 | 371 | 157 | 35 | 32 | 9 |
| 12 | 64 | 211 | 3 | 9 | 176 | 4 | 3 | 14 | 2 | 4 | 18 | - | 4 | 5 | 3 | 7 | 3 | 45 |
| 13 | 4 | 95 | 2 | 11 | 32 | 15 | 96 | 13 | 3 | 4 | 168 | 6 | - | 230 | 10 | 23 | 2 | 15 |
| 14 | 5 | 97 | 6 | 24 | 30 | 209 | 64 | 19 | 11 | 8 | 595 | 3 | 129 | - | 30 | 44 | 6 | 9 |
| 15 | 3 | 91 | 36 | 15 | 26 | 13 | 49 | 17 | 157 | 32 | 651 | 3 | 11 | 52 | - | 14 | 23 | 12 |
| 16 | 3 | 120 | 3 | 511 | 26 | 243 | 7 | 6 | 4 | 3 | 72 | 7 | 11 | 41 | 9 | - | 2 | 10 |
| 17 | 77 | 461 | 5 | 12 | 49 | 10 | 6 | 349 | 5 | 81 | 172 | 8 | 8 | 21 | 22 | 11 | - | 22 |
| 18 | 5 | 58 | 2 | 14 | 117 | 5 | 4 | 6 | 2 | 3 | 12 | 25 | 5 | 6 | 3 | 8 | 2 | - |

City/county: 1 = Changnyeong, 2 = Changwon, 3 = Geochang, 4 = Geoje, 5 = Gimhae, 6 = Goseong, 7 = Hadong, 8 = Haman, 9 = Hamyang, 10 = Hapcheon, 11 = Jinju, 12 = Miryang, 13 = Namhae, 14 = Sacheon, 15 = Sancheong, 16 = Tongyeong, 17 = Uiryeong, 18 = Yangsan. The value of P.F./100 L.P: the daily number of the population who travel from the origin to destination per origin's 100 local population.

4.2.2. Network Analysis

The network analysis was performed using the internal network of eighteen areas in SGP. As a result, Changwon and Jinju were ranked in the top 20% in both indicators of EC and FBC (Table 5, Figure 5). As aforementioned, in the value of P.F./100 L.P., Changwon and Jinju had outstanding inflow patterns from the neighboring areas. Furthermore, the two cities were positioned in the high hierarchy and acted as mediators in the internal networks of SGP. Following the outcome, Changwon and Jinju were proposed as central cities of the internal networks of SGP.

**Table 5.** Results of network analysis of the internal flow of SGP.

| No | City/County | EC | City/County | FBC |
|---|---|---|---|---|
| 1 | Changwon | 0.622 | Jinju | 21,839 |
| 2 | Haman | 0.541 | Changwon | 21,647 |
| 3 | Uiryeong | 0.263 | Tongyeong | 14,591 |
| 4 | Jinju | 0.216 | Sacheon | 13,678 |
| 5 | Gimhae | 0.213 | Goseong | 13,615 |
| 6 | Changnyeong | 0.192 | Haman | 12,012 |
| 7 | Goseong | 0.185 | Sancheong | 11,073 |
| 8 | Sacheon | 0.146 | Gimhae | 11,006 |
| 9 | Sancheong | 0.118 | Hamyang | 10,409 |
| ... | ... | ... | | ... |
| 18 | Geochang | 0.040 | Yangsan | 3921 |
| 19 | Miryang | 0.096 | Hapcheon | 9485 |
| 20 | Hadong | 0.085 | Changnyeong | 9011 |
| Concentric map [1] | 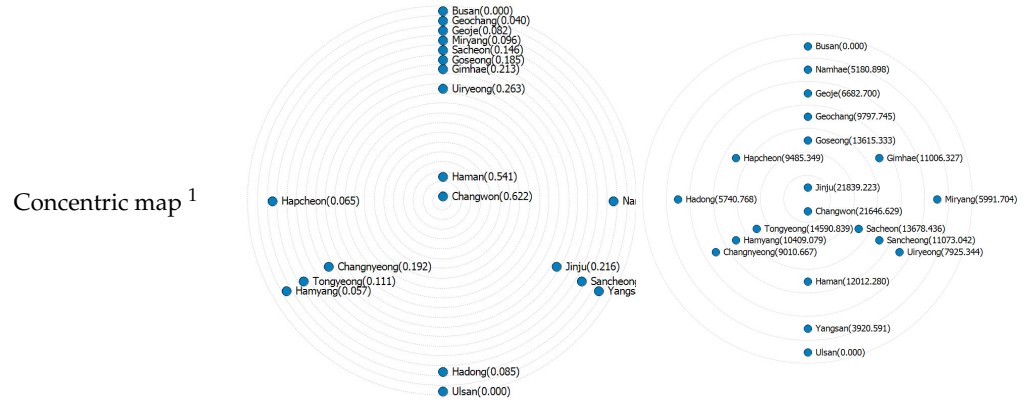 | | | |

[1] In the concentric map, an area with a high value of centrality was located at the center of the circle.

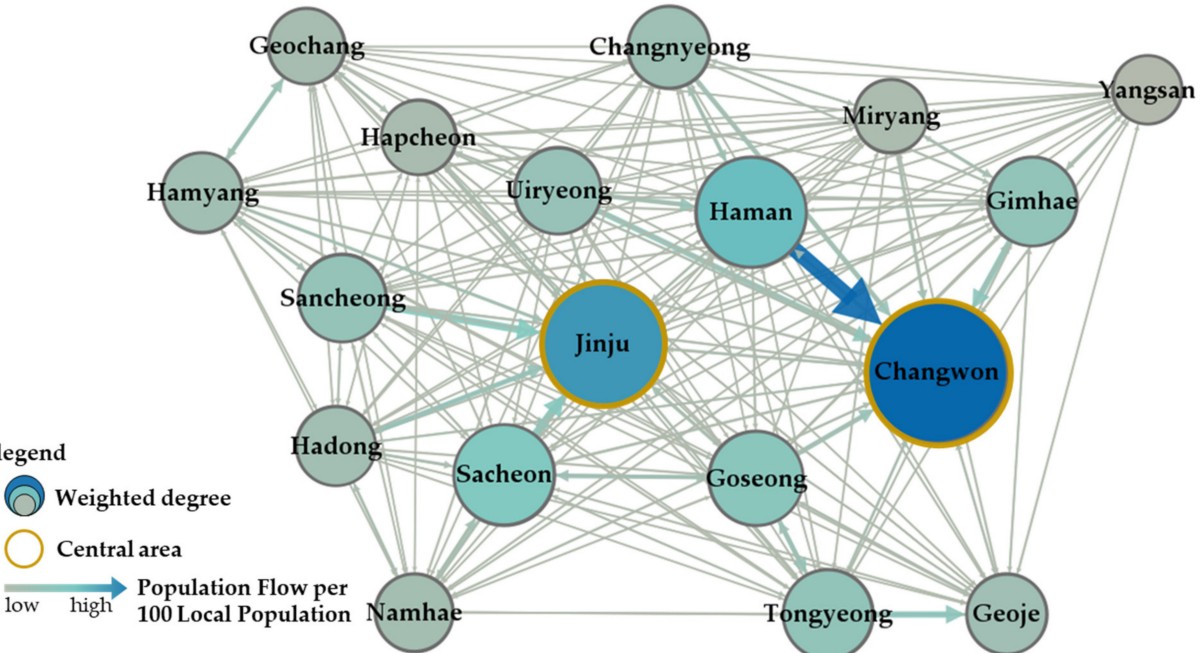

**Figure 5.** Urban networks and central areas of the internal flow of SGP.

In Figure 5, an area that had the higher value of centrality had a larger and darker circle. The arrows represented "population flow per 100 local population" (P.F./100 L.P.),

which had the higher value, it had thicker and darker arrows. The pointing of arrows indicated the flow direction of the value.

### 4.3. Inconsistency of Central Cities in In/External Networks of SGP

This research defined the central cities in both internal/external networks of SGP; as a result, Yangsan, Gimhae, and Geoje were observed in the external network, while Changwon and Jinju were identified in the internal network. It demonstrated that the central areas differed from the internal and external networks of SGP. This outcome stated that the current urban networks within SGP, Busan, and Ulsan were hardly linked horizontally and complementary connected. Moreover, the significant flows and interactions in both internal and external networks mostly rely on geographical closeness. Further, most of the areas of SGP, which were not closely located, had a low degree of connectivity with Busan and Ulsan. Changwon and Jinju were identified as the central cities in the internal network, nevertheless, these two cities seldomly functioned as central areas in Busan and Ulsan. These results clearly showed that the current spatial structure of SGP would be explained as the vertical structure mainly between Changwon and Jinju, with high positions in the hierarchy, and several spatially adjacent areas with them. Additionally, it implied that most counties in SGP, which were located far from the two central cities, were scarcely interconnected with the two central areas and were isolated.

The above results indicated that it is necessary to discuss a strategy for interconnecting the areas of the BUG megacity to make the megacity function as one urban region. In this study, we aimed to suggest a first step towards encouraging a fully connected BUG megacity. In this context, we investigated spatial strategy focusing on SGP to develop multifarious relations within areas of SGP. Therefore, this paper sought to identify the spatial structure of SGP by defining the primary and secondary centers and proposed some suggestions to interconnect these central areas. In the previous Section 4.2, we conducted the network analysis of the internal network of SGP and postulated the central cities as Changwon and Jinju. However, only the two cities considered as central areas would be difficult to realize a properly connected spatial structure of SGP. Therefore, we defined Changwon and Jinju as the primary centers of SGP. Meanwhile, there were no second-hierarchy central areas; thus, this study investigated secondary centers by conducting community detection and network analysis in the following section.

### 4.4. Examining Secondary Centers of Communities in SGP
#### 4.4.1. Community Detection

To identify communities within SGP, we conducted community detection using the internal network of SGP. As a result, four communities were investigated (Table 6). Firstly, community 1 included Changwon, Gimhae, Miryang, Yangsan, Uiryeong, Changnyeong, and Haman. In this community, Changwon had the most outstanding networks with the other areas. Among them, Gimhae, Miryang, and Yangsan had strong urban networks with Busan and Ulsan, and the three counties, Uiryeong, Changnyeong, and Haman, were closely located with Changwon. Secondly, community 2 contained Jinju, Sacheon, Sancheong, Hadong, and Namhae. Jinju had the most significant interaction with the other areas of the community. In particular, the strong inflows to Jinju from Sacheon, Sancheong, and Hadong were especially noticeable. Thirdly, community 3 included Geoje, Tongyeong, and Goseong. It was considered that, since Geoje and Tongyeong were coastal cities, the two cities had strong relations with Goseong. Fourthly, community 4 contained Geochang, Hamyang, and Hapcheon. This community tended to be isolated in SGP because all of the counties lacked a well-organized transportation system to access the primary centers, Changwon and Jinju.

**Table 6.** Results of community detection and defining central cities.

| Community | No | City/County | EC | FBC | Central Cities The First | Central Cities The Second |
|---|---|---|---|---|---|---|
| 1 | 1 | Changwon | 0.661 | 3178 | V | |
| | 2 | Haman | 0.609 | 2724 | | V |
| | 3 | Changnyeong | 0.218 | 2023 | | |
| | 4 | Gimhae | 0.229 | 1999 | | |
| | 5 | Miryang | 0.108 | 1478 | | |
| | 6 | Uiryeong | 0.282 | 961 | | |
| | 7 | Yangsan | 0.046 | 691 | | |
| 2 | 1 | Jinju | 0.661 | 2220 | V | |
| | 2 | Sacheon | 0.466 | 1149 | | V |
| | 3 | Hadong | 0.322 | 1043 | | |
| | 4 | Namhae | 0.234 | 829 | | |
| | 5 | Sancheong | 0.433 | 439 | | |
| 3 | 1 | Tongyeong | 0.685 | 0.685 | | V |
| | 2 | Geoje | 0.581 | 0.581 | | |
| | 3 | Goseong | 0.440 | 0.440 | | |
| 4 | 1 | Geochang | 0.702 | 209 | | V |
| | 2 | Hamyang | 0.661 | 22 | | |
| | 3 | Hapcheon | 0.265 | 22 | | |

### 4.4.2. Secondary Centers of Four Communities

An area's centrality was defined by the extent of urban networks. Specifically, depending on how many links and nodes interacted with the area, its centrality was evaluated. It implied that the centrality of a certain area in SGP could be differently defined by setting the scope of the network as SGP or community in SGP. Therefore, this study performed a network analysis of four communities of SGP to examine the central areas of every community. Based on the results of the network analysis, the area having the highest values in both EC and FBC was deemed as the secondary center; ranked in the top 20% among eighteen areas based on the result of network analysis. As aforementioned, Changwon and Jinju were defined as the primary centers; thus, the two areas were excluded when examining secondary centers. The results of identifying the secondary centers of each community are as follows (Figure 6, Table 6).

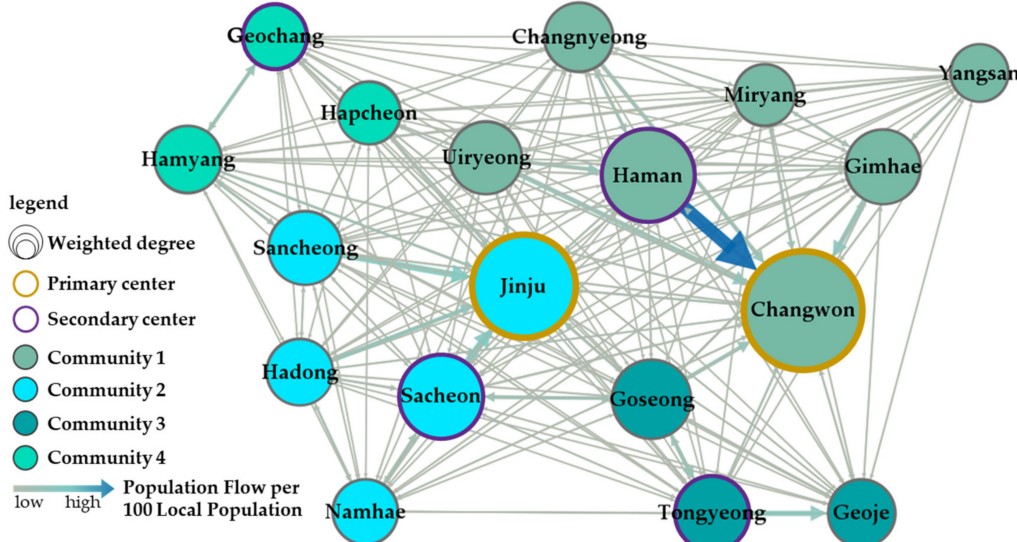

**Figure 6.** Primary and secondary centers in communities in SGP.

## 5. Discussion

### 5.1. Measures to Enhance Bidirectional Urban Networks

The growth of megacities is a significant global trend. Megacities include diverse cities of different functions and sizes. Identifying and optimizing the spatial structure of megacities concerning their functions and sizes is a vital issue in balanced spatial development. In addition, the spatial structure of the city network model emphasizes bidirectional connections within areas in a megacity. Nevertheless, this study found asymmetries in urban connectivity and imbalances in the directions and sizes of the weighted flows across cities and counties in the BUG megacity. These results are consistent with existing findings that several megacities demonstrate unidirectional and hierarchical linkages in their urban networks and marginalize small cities and counties [95–97]. Accordingly, cross-regional spatial development strategies that consider small- and medium-sized cities are necessary [97]. In this vein, many studies have argued that integrated multi-level public transport systems can strengthen horizontal urban connectivity in megacities [98,99]. For example, Wang and Lu [100] showed that a transportation network contributed to increases in both the number of tourists and tourism profits in China; specifically, they pointed out that accessibility had a spatial-temporal contraction effect on megacities and prefecture-level cities. Meanwhile, Hou [101] advised that designs for future public traffic systems should focus on improving inter-city networks between higher-order cities and lower-order cities. Additionally, Chung et al. [102] studied spatial strategies for urban regions to develop a spatial structure for the network city model and proposed strategies regarding transportation and land use in megacity/urban regions (Figure 7). Firstly, focusing on the primary centers, Chung et al. recommended a rapid transit system based on the transit-oriented development (TOD) model. While the traditional TOD model was adopted in transfer stations in urban areas, recent theoretical studies have applied this model in urban corridors and/or urban regions. Secondly, Chung et al. suggested the implementation of a linked transit system between primary and secondary centers based on the transit-oriented corridors (TOC) model. Thirdly, they also advised the creation of a cyclical transit system among secondary centers based on the TOC model [102].

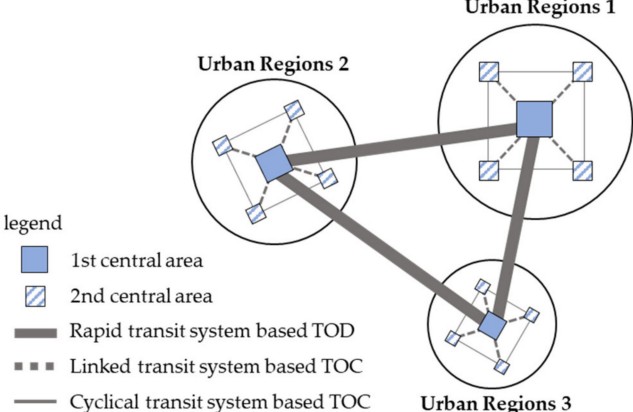

**Figure 7.** Strategies to construct a network urban spatial structure. Source: adapted from Chung et al. [102] (p. 75).

### 5.2. Enhancing Vertical and Horizontal Transportation Networks across Primary and Secondary Centers

This section considers the major plans and projects regarding the public transportation system in the BUG megacity. A Korean high-speed rail project was launched in 2019, and a high-speed railway network connecting Gimcheon–Geoje will be created in 2027. This train will travel from Geoje to Seoul in less than three hours. The old western part of SGP has not historically had access to transportation services; therefore, the high-speed railway network will increase accessibility from SGP to Seoul [103,104]. Moreover, a double-track electric/express railway is currently under construction that stands to improve accessibility

among the central cities of the BUG megacity, including Jinju, Changwon, Busan, and Ulsan; this train will travel from Jinju to Ulsan in an hour [105]. These currently discussed high-speed railway networks were illustrated as black lines in Figure 8. Notably, the abovementioned projects were designed to connect the areas of the BUG megacity across the east–west axis; this highlights the desire to improve horizontal transportation networks in the region by connecting the four main central cities (i.e., Changwon, Jinju, Busan, and Ulsan).

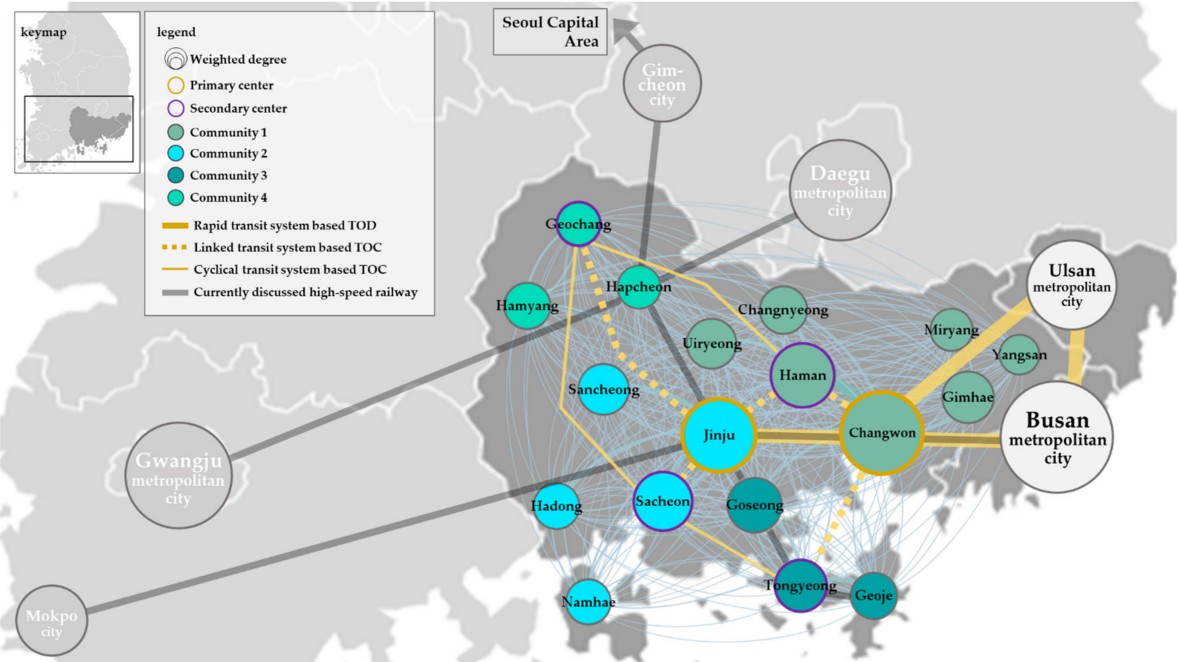

**Figure 8.** Strategies to strengthen networks of the BUG megacity focusing on primary/secondary centers.

In this vein, we turn to the study by Chung et al. [102] to consider a spatial strategy for connecting the areas of the BUG megacity and developing the megacity as one urban region. Firstly, this study identified the primary centers of SGP as Changwon and Jinju. At the same time, the two primary centers were defined as the central cities in the BUG megacity in its spatial structure plan. Therefore, the rapid transit system based on the TOD model should extend across these cities in ways that integrate neighboring small- and medium-sized cities and counties in SGP. Put differently, the spatial plans should include a rapid transit system based on the TOD model that centers on Changwon and Jinju.

Meanwhile, existing practical strategies for strengthening the network generally fail to adequately consider the secondary centers and rural areas in SGP; better plans must be made to improve accessibility between the major centers and small- and medium-sized cities and counties. Therefore, a linked transit system based on the TOC model should be created between the secondary and primary centers of Haman–Changwon, Tongyeong–Changwon, Sacheon–Jinju, and Geochang–Jinju. This will improve the networks across the primary centers and enhance vertical accessibility across the SGP. Moreover, a cyclical transit system based on the TOC model should also be considered for the secondary centers, such as Haman–Geochang–Sacheon–Tongyeong. A cyclical system would improve vertical accessibility within the counties and small- and medium-sized cities in SGP. Over the long term, northern communities in the SGP, including Geochang, Hacheon, and Hamyang, may be used to connect the BUG megacity with other neighboring urban regions, such as Daegu, Gwangju, Yeosu, and Suncheon. In summary, a systemic transportation network that extends vertically and horizontally across the SGP is needed. Figure 8 presents the above strategies for creating such vertically and horizontally accessible urban networks across the primary and secondary centers in the region, which were illustrated as the yellow

lines. These suggestions could serve as lessons for planning multi-directional connections within areas in megacities.

Notably, Kuo et al. [106] surveyed recent studies and technologies on smart public transport systems, such as feeder services [107–110], multi-modal transport networks [111,112], and integrated passenger routing [113,114], and underscored that the integrated multi-modal public transport system provides proper service coverage and frequency. Urban planners and policymakers should consider such emerging studies and technologies in formulating spatial development plans for megacities.

### 5.3. Limitations

Beyond its empirical contributions, this study presents some limitations. Firstly, the geographical scope of the research included the BUG megacity; however, the study did not include population flow data from Busan to Ulsan. Secondly, when determining the central areas, this study only considered the value of centrality and did not investigate the urban industry and economic indices. Therefore, in future research, more abundant databases should be addressed to more accurately identify the central cities and spatial structures of BUG megacities. Thirdly, the central areas were analyzed at the city and/or county level. Future studies should examine urban networks both at the city and/or county level and neighborhood level to identify the spatial structure in a multidimensional way. Despite these limitations, this study's findings yielded the insight that primary and secondary centers should be built across areas in SGP as well as the BUG megacity.

**Author Contributions:** Y.B. studied the literature reviews and wrote the manuscript. Y.B. performed data curation, formal analysis, and visualization. H.J. contributed to recommendations on the analytical framework and supervised the development of the manuscript as the corresponding author. H.J. addressed funding acquisition and provided the population flow data. All authors contributed to modifying this manuscript. All authors have read and agreed to the published version of the manuscript.

**Funding:** This work was supported by the National Research Foundation of Korea (NRF) grant funded by the Korean Government (MSIT, No. 2020R1G1A1101214).

**Institutional Review Board Statement:** Not applicable.

**Informed Consent Statement:** Informed consent was obtained from all subjects involved in the study.

**Data Availability Statement:** The data presented in this study are available on request from the corresponding author. The data are not publicly available due to containing personal information.

**Acknowledgments:** The authors thank the anonymous reviewers and editors for their valuable and constructive suggestions for improving this article.

**Conflicts of Interest:** The authors declare no conflict of interest.

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
