# Peer review of "A Study on the Spatial Structure of the Bu-Ul-Gyeong Megacity Using the City Network Paradigm"

_sustainability, doi:10.3390/su142315845_

Round 1
Reviewer 1 Report
I suggest to avoid a large number of shortcuts in Abstract. Full names are OK, but shortcuts could be explained in Introduction (like in Abstract now)
line 62: "network city theory" which one? reference needed
Introduction part should be reorganized. Is should contain previous works/theories and the goal of the paper with information what new authors introduce to current knowledge. Now literature review in Introduction is quite poor. Also informations about methods, and generel findings should be moved to other chapters. BUG overview started at line 39 should be moved to Ch. 3.1
Chapter 2.4 should be moved to Introduction
Some basic map of Korea with BUG boundaries could be added. It could help for international readers, who don't know Korea well.
Mathematic formulas used by authors should be added (with explanation) in Chapter 3
The paper is about Korean megacity, but Sustainability is the international Journal. In this case I recommend to add some sentences about possibilities of use author's methods/findings in other countries.
Reviewer 2 Report
Dear Authors,
The proposed study provides a thorough discourse on spatial knowledge of cities through the lens of city networks. In this regard, this paper presents an interesting study and method in this area.
I globally have some comments to give a general overview according to the following points:
1) In the introduction part, the paper presented explains the basic overview and reasoning of the investigation. However, in addition to the contents mentioned, I believe the introduction should conclude with information about the next sections.
2) Some figures are blurred and difficult to understand (especially see figure 4). Furthermore, a brief description of these figures is beneficial in understanding their context in the study.
3) Discussion area may use some major improvement. The duration of the network analysis should be included in the methods section rather than the discussion section. Additionally, this section should offer a critical evaluation of your analysis, which currently seems unconvincing. Furthermore, this section should incorporate a discussion of the assessed analysis. Emphasize how the broader city will benefit from the proposed strategy.
4) The paper needs additional using updated references.
Reviewer 3 Report
The manuscript deals with a very pertinent and interesting topic related to the spatial structure of the Bu-Ul-Gyeong megacity in city network paradigm. The following points however, need to be addressed, before the manuscript can be recommended for final publication:
Comment1: In my opinion, the abstract is not perfectly written. The abstract should be reworked to be more conscious and influential. Furthermore, no quantitative results are presented in the abstracts. many readerships may only read the abstract of your work. Hence, the abstract must be able to stand alone. Mostly, the abstract is the only part of the research article that will be accessible in indexing databases; getting a good impression is very important.
Comment2: The methodology is suggested to corresponds to the results.
Comment3: The conclusion must be added and it is hoped to be concise and brief.
Comment4: Page 2, Line: 47: you use lump sum references which are not recommended. Please, avoid using lump sum references as much as possible.
Comment5: Several graphs must be of sufficient clarity, such as Fig. 4, Fig. 5, etc al.
Comment6: There are only 16 x references from 2022. At this time where 2022 is almost over, there could be some more references from the latest research / literature.
Round 2
Reviewer 1 Report
I accept the paper in present form.